# Anti-Disturbance Compensation-Based Nonlinear Control for a Class of MIMO Uncertain Nonlinear Systems

**DOI:** 10.3390/e23111487

**Published:** 2021-11-10

**Authors:** Wameedh Riyadh Abdul-Adheem, Ahmed Alkhayyat, Ammar K. Al Mhdawi, Nik Bessis, Ibraheem Kasim Ibraheem, Ahmed Ibraheem Abdulkareem, Amjad J. Humaidi, Arif A. AL-Qassar

**Affiliations:** 1Department of Electrical Engineering, College of Engineering, University of Baghdad, Baghdad 10001, Iraq; wameedh.r@coeng.uobaghdad.edu.iq (W.R.A.-A.); ibraheemki@coeng.uobaghdad.edu.iq (I.K.I.); 2College of Technical Engineering, The Islamic University, Najaf 54001, Iraq; ahmedalkhayyat85@iunajaf.edu.iq; 3Department of Computer Science, Edge Hill University, Ormskirk L39 4QP, UK; Nik.Bessis@edgehill.ac.uk; 4Department of Computer Engineering Techniques, Al-Rasheed University College, Baghdad 10001, Iraq; 5Control and Systems Engineering Department, University of Technology, Baghdad 10001, Iraq; 60162@uotechnology.edu.iq (A.I.A.); amjad.j.humaidi@uotechnology.edu.iq (A.J.H.); 60023@uotechnology.edu.iq (A.A.A.-Q.)

**Keywords:** MIMO systems, active disturbance rejection control, decentralized control, nonlinear control, subsystem couplings, extended state observer, uncertainties, output tracking

## Abstract

Multi-Inputs-Multi-Outputs (MIMO) systems are recognized mainly in industrial applications with both input and state couplings, and uncertainties. The essential principle to deal with such difficulties is to eliminate the input couplings, then estimate the remaining issues in real-time, followed by an elimination process from the input channels. These difficulties are resolved in this research paper, where a decentralized control scheme is suggested using an Improved Active Disturbance Rejection Control (IADRC) configuration. A theoretical analysis using a state-space eigenvalue test followed by numerical simulations on a general uncertain nonlinear highly coupled MIMO system validated the effectiveness of the proposed control scheme in controlling such MIMO systems. Time-domain comparisons with the Conventional Active Disturbance Rejection Control (CADRC)-based decentralizing control scheme are also included.

## 1. Introduction

In the control discipline, some systems are MIMO in their nature; indeed, the control theories for such systems will notice direct applications in a wide assortment of fields, such as space innovation, electric machines, and robotic control. The control of MIMO systems is a challenging task because of the state and input couplings. Moreover, if the MIMO systems are uncertain and nonlinear, then the control task turns out to be more challenging. In this respect, theoretical outcomes and beneficial practices for structuring satisfactory controllers are tremendously scarce.

In recent years, many researchers have discovered the aforementioned challenges and the exploration for the solutions to these problems using different techniques, such as fuzzy logic [1,2,3,4,5,6], neural networks [7,8,9], and sliding mode techniques [10,11,12,13,14,15,16,17]. In [18], a novel decentralized optimal control strategy was developed using the online learning of neural networks to stabilize a class of continuous-time nonlinear interconnected large-scale systems. A linear periodic controller with decentralized and centralized settings that provide linear quadratic regulator (LQR) optimal performance is demonstrated in [19]. While in [20], a decentralized controller with the controllability of a linear time-invariant (LTI) system has been discussed. With this technique, a measure-based Hankel operator was developed, which gave rise to a measure that combined the controllability Gramian, observability Gramian, and cross-Gramian incorporating the information structure. The work of [21] investigated a design methodology for the decentralized voltage controllers that act on the distributed generation reactive power injections. A decentralized control scheme was reported in the literature to control the reactive and active power of a grid-tied AC-stacked PV inverter architecture using single-member phase compensation [22]. On the other hand, distribution networks have been amalgamating grid-connected photovoltaic (PV) systems, which can actively contribute to flatten the voltage profile of the feeders by injecting reactive power. Nevertheless, the contemporary operation of the PV system might cause difficulties in regulating voltage and stability. To counteract these problems, the work in [23] pointed out a design method for a decentralized voltage controller to adjust the reactive power injection of the PV. The decentralized control problem solved in [24] is based on a policy iteration algorithm for large-scale nonlinear systems with unknown mismatched interconnections. The work in [25] outlined the application of the decentralized approach for controlling and coordinating the Autonomous Guided Vehicles (AGV) system. In [26], a recursive decentralized controller has been proposed for the motion control of space manipulators, where the space manipulator is considered as a group of distinct second-order systems. The control signal depends only on the joint measurements in each subsystem. The authors of [27] suggested a new fractional decentralized control for aircraft engines, which are considered as uncertain large-scale systems composed of interconnected uncertain subsystems. Finally, the authors in [28] validated a dynamic inversion technique in combination with a disturbance estimator and applied it to MIMO nonlinear systems.

In this paper, a control scheme is proposed based on the decentralized principle in which the input couplings for the uncertain nonlinear MIMO system is first resolved, converting it into decoupled Single-Input-Single-Output (SISO) linear time-invariant systems, then followed by an application of an IADRC for each of the SISO systems separately. This technique has the advantage of reducing model dependence in its design as compared to the aforementioned works [3,4,5,6,7,8,9,10,11,12,13,14,15,16,17]. The suggested IADRC-based decentralized control configuration does not require a huge tuning to its coefficients similar to the adaptive control methods that are based on neural networks. Furthermore, common shortcomings such as chattering in sliding mode control techniques are circumvented in the suggested IADRC-based configuration. Finally, the suggested control configuration is a real-time strategy, which implies it observes/eliminates the estimated total disturbance in an online fashion without a requirement for a choice from a reasoning engine as in fuzzy logic, where vast fuzzy logic rules must be designed and saved in a database.

The contributions of the paper are elucidated as given next. An IADRC-based decentralized control scheme is proposed by refining the dynamic interactions between different subsystems into the generalized disturbance for later estimation/cancelation from the input channel through a feedback control law based on a novel Extended State Observer (ESO). The classical ADRC configuration is enhanced by two modifications, firstly, adding a new nonlinear error function to the error-correcting term of the classical ESO to increase the sensitivity of the observer estimation to the small changes in the estimation errors. Secondly, a more augmented state is added to the dynamics of the classical ESO to estimate the generalized disturbances with higher-order derivatives. Thus, the new observer is called a Nonlinear Higher Order ESO (NHOESO).

This paper is structured as follows. Section 2 presents the problem statement followed by the succinct introduction on the Active Disturbance Rejection Control and the formulation of the generalized disturbance in Section 3. Section 4 introduces the proposed IADRC-based decentralized control scheme, the IADRC configuration, and the stability analysis of the closed-loop system using Hurwitz stability. Section 5 demonstrates the numerical simulations of the proposed IADRC-based decentralized control scheme on a hypothetical highly nonlinear MIMO system. Section 6 concludes the paper.

## 2. Statement of the Problem

Given a nonlinear MIMO system given as
(1){ξi(γi)=fi(ξ,η,w)+∑j=1pgi,j(t)uj, yi=ξi, i∈{1,2,…,p}.  
where y=(y1(t),y2(t),…yp(t))T∈ℝp is the output of the MIMO system, u=(u1(t),u2(t),…up(t))T∈ℝp is the input, fi∈C(ℝ γ×(n−γ)×p, ℝ), i∈{1,2,…,p} is an unknown system function, ξ=(ξ1(t),ξ2(t),…,ξp(t))T∈ℝγ is the state vector, w=(w1(t),w2(t),…,wp(t))T∈ℝp is the external disturbance, and gi,j∈C(ℝ, ℝ) is an unknown gain function. The system of (1) includes internal dynamics, which can be described as η˙=f0(ξ, η,w), where f0∈C(ℝ γ×(n−γ)×p,ℝ(n−γ)) is an unknown internal dynamic.

Consider the *i*-th subsystem with state vector named as ξi(t)=(ξi(t),…,ξi(γi−1)(t))T∈ℝγi, i∈{1,2,…,p}. The coefficient bi,j is an approximation for gi,j in the system within a ±50% range [29,30], then (1) is rewritten as,
(2){ξi(γi)=fi(ξ, η,w)+∑j=1p(gi,j(t)−bi,j)uj+∑j=1pbi,juj yi=ξi, i∈{1,2,…,p} 
(3)Fi=fi(ξ, η,w)+∑j=1p(gi,j(t)−bi,j)uj, i∈{1,2,…,p}
where Fi is the generalized disturbance of the MIMO nonlinear system (1)

Figure 1 depicts nonlinear system of (2) with the generalized disturbance of (3). It is required to design an active disturbance rejection control-based nonlinear controller for the multi-input-multi-output (MIMO) system of (1) such that the following objectives are satisfied, dissociation of the couplings between the states, dissociation of the couplings between different inputs, canceling the effect of the generalized or total disturbance $F_{i}$, i∈{1,2,…,p} on the system’s output, and preserving a satisfactory performance throughout both the steady-state and transient.

## 3. Theoretical Background

The basic principle of the Active Disturbance Rejection Control (ADRC) lies in estimating in a real-time manner the system dynamics along with “*generalized disturbance*” [31], which includes all the undesirable system uncertainties and external disturbance by using ESO. The ESO is the main unit of the active disturbance rejection control (ADRC) [32]. The ADRC includes an ESO, a Nonlinear State Error feedback (NLSEF), and a Tracking Differentiator (TD) as shown in Figure 2. where r∈ℝ is the reference signal, (r1r2…rρ)T∈ℝn is the transient profile, ρ is the relative degree, b0 is the gain of the control input, (ξ^1ξ^2…ξ^ρ+1)T∈ℝn+1 is the extended observed vector which involves the predicted generalized disturbance ξ^ρ+1 and predicted states ξ^1,…,ξ^ρ of the system, and v∈ℝ is the control input. Tracking Differentiator (TD) which is used to generate the transient profile of the reference input (i.e., the noise-free signal itself together with its ρ − 1 derivatives).

Several engineering control application tasks have been effectively fixed in the last two decades, via the effective implementation of ADRC. These include flexible-joint manipulator control [33], omnidirectional mobile robot control [34], aerospace [35], temperature control [36], DC-DC power converters [37], speed control of permanent magnet DC motor [38], control of power output of wind turbines [39], Energy Storage Grid-Connected Inverter [40], Lower Limb Exoskeleton in Swing Phase [41], Ship Dynamic Positioning Systems [42], Vibration Suppression in Position Servo Systems [43], speed control of Differential drive mobile robot [44], Hydraulic Valve-Controlled Hydraulic Motor [45], the applications of ADRC on unmanned aerial vehicles are highlighted in [46,47,48,49].

## 4. Main Results

In this section, we present the proposed IADRC-based decentralized control scheme. Then, the IADRC is designed to apply the required functionalities of the uncertainty and disturbance elimination for the MIMO nonlinear uncertain systems. The stability of the closed-loop system including the IADRC is established.

### 4.1. The Proposed IADRC-Based Decentralized Control Scheme

In this IADRC-Based Decentralized Control Scheme, the nonlinear-coupled MIMO system (1) is altered into a set of SISO systems by including the coupling inputs into the generalized disturbance. The MIMO system given in (1) is rewritten as given in Equation (4) for i∈{1,2,…,p},
(4)$$\{\begin{array}{c} \xi_{i}^{\left(\gamma_{i}\right)}=f_{i}\left(\xi ,\eta ,w\right)+\overset{p}{\underset{j=1}{\sum }}\left(g_{i,j}\left(t\right)-b_{i,j}\right)u_{j}+\overset{p}{\underset{j=1,j\neq i}{\sum }}b_{i,j}u_{j}+b_{i,i}u_{i},\\ y_{i}=\xi_{i},\; \end{array}$$

The coupling inputs is included in the generalized disturbance $F_{i}^{\prime }$ which is expressed as,
(5)$$F_{i}^{\prime }=f_{i}\left(\xi ,\eta ,w\right)+\overset{p}{\underset{j=1}{\sum }}\left(g_{i,j}\left(t\right)-b_{i,j}\right)u_{j}+\overset{p}{\underset{j=1,j\neq i}{\sum }}b_{i,j}u_{j}$$

Finally, the MIMO nonlinear system is written in a simple form given as,
(6a)$$\{\begin{array}{c} \xi_{i}^{\left(\gamma_{i}\right)}=F_{i}^{\prime }+b_{i,i}u_{i}\;,\;\\ y_{i}=\xi_{i},\;i\in \left\{1,2,\ldots ,p\right\}\; \end{array}$$

By expanding the $\gamma_{i}$-th derivative of ξ in Equation (6a) into a set of state-space equations of size $\gamma_{i}$, Equation (6a) can be converted into (6b) as a chain of integrators. Let $\xi_{i,l}=\xi_{i}^{\left(l-1\right)}$, $l\in \left\{1,2,\ldots ,\gamma_{i}\right\},\;i\in \left\{1,2,\ldots ,p\right\}$, i.e., $\xi_{i,1}=\xi_{i}^{},\;\xi_{i,2}={\dot{\xi }}_{i}^{},\;\xi_{i,3}={\ddot{\xi }}_{i}^{},\;\ldots .etc.$ Furthermore, assume $\xi_{i,\gamma_{i}+1}=F_{i}^{\prime }\Rightarrow {\dot{\xi }}_{i,\gamma_{i}+1}=\dot{F_{i}^{\prime }}$. The subsystem (6a) can be written as,
(6b)$$\{\begin{array}{c} {\dot{\xi }}_{i,1}=\xi_{i,2},\;\\ \begin{array}{c} {\dot{\xi }}_{i,2}=\xi_{i,3},\;\\ \vdots \;\\ {\dot{\xi }}_{i,\gamma_{i}}=F_{i}^{\prime }+b_{i,i}u_{i},\; \end{array}\;\\ {\dot{\xi }}_{i,\gamma_{i}+1}={\dot{F}}_{i},\;i\in \left\{1,2,\ldots ,p\right\} \end{array}$$

The system of (6) and the considered generalized disturbance $F_{i}^{\prime }$ of (5) is illustrated in Figure 3. The IADRC-based decentralized scheme for controlling the nonlinear MIMO system (6) is shown in Figure 4.

There are two methods with which to select the value of the coefficient $b_{i,i}\in R\backslash \left\{0\right\},\;i\in \left\{1,2,\ldots p\right\}$:(i)A rough estimate of *b* (*t*) in the system within a ±50% range [30].(ii)Typically it is selected perspicuously by the designer as a design parameter [50].

### 4.2. The Configuration of the Improved ADRC (IADRC)

The proposed control arrangement to control the MIMO nonlinear system given in (4) includes a Nonlinear Higher Order ESO (NHOESO) that replaces the conventional Linear ESO (LESO) utilized by the CADRC. The dynamic state-space representation of the Tracking Differentiator (TD) can be expressed as [30],
(7)$$\{\begin{array}{c} {\dot{r}}_{i,l}=r_{i,l+1},\;l\in \left\{1,2,\ldots ,\gamma_{i}-1\right\}\;\\ {\dot{r}}_{i,\gamma_{i}}=-R_{i}sign\left(r_{i,1}-r_{i}+\frac{r_{i,2}\left|r_{i,2}\right|}{2R_{i}}\right),\;i\in \left\{1,\ldots ,\;p\right\}\; \end{array}$$
where $R_{i}$, i∈{1,…, p}, is an application-dependent design parameter, and its value controls the convergence speed of the differentiator output. The NLSEF has a nonlinear error function given as follows [30],
(8)$$fal_{i,l}\left({\tilde{e}}_{i,l},\alpha_{i,l},\delta_{i,l}\right)=\{\begin{array}{cc} \frac{{\tilde{e}}_{i,l}}{\delta^{1-\alpha_{i,l}}}\; & \left|{\tilde{e}}_{i,l}\right|\leq \delta_{i,l}\;\\ {\left|{\tilde{e}}_{i,l}\right|}^{\alpha_{i,l}}sign\left({\tilde{e}}_{i,l}\right) & \left|{\tilde{e}}_{i,l}\right|\geq \delta_{i,l} \end{array}$$
with $l\in \left\{1,2,\ldots ,\gamma_{i}\right\}$, i∈{1,…, p}, $\alpha_{i,l},\delta_{i,l}$ are design parameters, usually, $\delta_{i,l}$ is a small number and $\alpha_{i,l}\in \left(0,1\right)$. With a suitable choice for the values of these parameters, the error ${\tilde{e}}_{i,l}$ approaches zero in a very short time. The proposed nonlinear higher-order ESO (NHOESO) is given as,
(9)$$\left\{ \begin{array}{c} {\dot{\hat{\xi }}}_{i,l}={\hat{\xi }}_{i,l+1}+a_{i,l}\;\omega_{o,i}^{l}\;{ℊ}_{i}\left(y_{i}-{\hat{\xi }}_{i,1}\right)\;,\;l\in \left\{1,2,\ldots ,\gamma_{i}-1\right\},\\ {\dot{\hat{\xi }}}_{i,\gamma_{i}}={\hat{\xi }}_{i,\gamma_{i}+1}+a_{i,\gamma_{i}}\;\omega_{o,i}^{\gamma_{i}}\;{ℊ}_{i}\left(y_{i}-{\hat{\xi }}_{i,1}\right)+u_{i}^{*},\\ {\dot{\hat{\xi }}}_{i,\gamma_{i}+1}={\hat{\xi }}_{i,\gamma_{i}+2}+a_{i,\gamma_{i}+1}\;\omega_{o,i}^{\gamma_{i}+1}\;{ℊ}_{i}\left(y_{i}-{\hat{\xi }}_{i,1}\right),\;\\ {\dot{\hat{\xi }}}_{i,\gamma_{i}+2}=a_{i,\gamma_{i}+2}\;\omega_{o,i}^{\gamma_{i}+2}\;{ℊ}_{i}\left(y_{i}-{\hat{\xi }}_{i,1}\right),\;i\in \left\{1,\ldots ,\;p\right\}\; \end{array}\right.$$
where $\omega_{o,i}$ is the NHOESO bandwidth for the *i*-th subsystem to be tuned, the vector ${\left({\hat{\xi }}_{i,1},\ldots .,{\hat{\xi }}_{i,\gamma_{i}}\right)}^{T}$, i∈{1,…, p} are the estimated system model state and ${\hat{\xi }}_{i,\gamma_{i}+1},\;i\in \left\{1,\ldots ,\;p\right\}$ is the estimated generalized disturbance, $a_{i,s}$, $s\in \left\{1,2,\ldots ,\gamma_{i}+2\right\}$ is the associated design parameter, they are selected such that the following matrix is Hurwitz.
(10)$$E={\left[\begin{array}{ccccc} -a_{i,1} & 1 & 0 & \cdots & 0\\ -a_{i,2} & 0 & 1 & \cdots & 0\\ \cdots & \cdots & \cdots & \cdots & \cdots \\ -a_{i,\gamma_{i}+1} & 0 & 0 & \ddots & 1\\ -a_{i,\gamma_{i}+2} & 0 & 0 & \cdots & 0 \end{array}\right]}_{\left(\gamma_{i}+2\right)\left(\gamma_{i}+2\right)}$$

The function ℊi:ℝ→ℝ, i∈{1,…, p} is nonlinear and designed as in [51],
(11)ℊi(ei)=Ki,α|ei|αisign(ei)+Ki,β|ei|βiei
where Ki,α,Ki,β,αi and βi are positive design parameters, and ei is defined as
(12)ei=yi−ξ^i,1

### 4.3. Stability Analysis of the Closed-Loop System

The stability of the closed-loop system with the proposed IADRC-based decentralized control scheme is demonstrated next. A couple of Theorems are needed and adopted in the investigation of the system stability of the closed-loop; they are stated as follows.

**Assumption** **1.***There exist*Mi,h∈ℝ+*such that*supt∈[0,∞)|Δi,h(t)|=Mi,h, i∈{1,2,…,p}.

**Assumption** **2.**V:ℝρ+2→ℝ+*and*W:ℝρ+2→ℝ+*are continuously differentiable functions with:*(13)λ1‖η‖2≤V(η)≤λ2‖η‖2 , W(η)=‖η‖2(14)∑i=1ρ+1∂V(η)ηi(ηi+1−aik(η1ω0ρ)·η1)−∂V(η)∂yρ+2aρ+2k(η1ω0ρ)η1≤−W(η) 
where λ1 and λ2 are positive constants.

**Theorem** **1.***Given the system (6b), and the NHOESO (9), for any initial values*(15)$$\underset{t\to \infty }{\lim}\left|\xi_{i,l}-{\hat{\xi }}_{i,l}\right|=0,\;i\in \left\{1,2,\ldots ,p\right\},l\in \left\{1,\ldots .,\;\gamma_{i}\right\}$$
and
(16)$$\underset{t\to \infty }{\lim}\left|F_{i}^{\prime }-{\hat{\xi }}_{i,\gamma_{i}+1}\right|=0$$
where $\xi_{i}$, and ${\hat{\xi }}_{i}$ denote the solutions of (6b) and (9), respectively, i∈{1,2,…,ρ+2}.

**Proof.** See Appendix A. □ 

Moreover, the tracking differentiator given below is a reduced version of the tracking differentiator given in (7).
(17)$$\{\;\begin{array}{c} {\dot{r}}_{1}\left(t\right)=r_{2}\left(t\right),\;\\ {\dot{r}}_{2}\left(t\right)=-R^{2}\varphi \left(r_{1}\left(t\right)-r\left(t\right)\right)-Rr_{2}\left(t\right)\; \end{array}\;$$
which will be utilized in the next theorem.

**Theorem** **2.***Consider the dynamic system (17). If the signal* r(t)*is differentiable and*${sup}_{t\in \left[0,\infty \right)}\left|\dot{r}\left(t\right)\right|=B<\infty$*, then the solution of (17) is convergent in the sense that,*$r_{1}\left(t\right)\;$*is convergent to*r(t)*as*R→∞.

**Proof.** See Appendix A. □

In what follows, the closed-loop stability of the MIMO nonlinear system is investigated for the closed-loop system using the suggested IADRC-based decentralized control scheme with a generalized disturbance $F_{i}^{\prime }$.

**Theorem** **3.***Consider a nonlinear* n*-dimensional uncertain MIMO system of (6). If the augmented system of (6) is governed by a linearization control law *ui*defined as*(18)ui=vi−ξ^i, γi+1bi,i, i∈{1,…, p}*where*vi*is designed as,*(19)vi=𝓀i,1(e˜i,1)e˜i,1+…+𝓀i,l(e˜i,l)e˜i,l+…+𝓀i,γi(e˜i,γi)e˜i,γi*where* l∈{1,2,…,γi} and i∈{1,…, p}, 𝓀i,l: ℝ→ℝ+*is an even nonlinear gain function,*e˜i,l=ri,l−ξ^i,l , i∈{1,…, p}, l∈{1,2,…,γi}*are the closed-loop errors. Then, based on the outcomes of Theorems 1 and 2, the closed-loop system is asymptotically stable, i.e.,*limt→∞|e˜i,l|=0.

**Proof.** The tracking error ${\tilde{e}}_{i,l}$ of the closed-loop system is the difference between the estimated states ${\hat{\xi }}_{i,l}$ of the nonlinear system and the TD output $r_{i,l}$ can be described as
(20)$${\tilde{e}}_{i,l}=r_{i,l}-{\hat{\xi }}_{i,l}\;,\;i\in \left\{1,\ldots ,\;p\right\},\;l\in \left\{1,2,\ldots ,\gamma_{i}\right\}$$With Assumptions 1 and 2 hold, the tracking error ${\tilde{e}}_{i,j}$ is expressed as
(21)$${\tilde{e}}_{i,l}=r_{i}^{\left(l-1\right)}{}^{}-\xi_{i,l}\;,\;$$For the system in (6), the states $\xi_{i,l}$ are defined as a function of the output of the system,
(22)$$\xi_{i,l}=y_{i}^{\left(l-1\right)}{}^{}\;,\;i\in \left\{1,\ldots ,\;p\right\},\;l\in \left\{1,2,\ldots ,\gamma_{i}\right\}$$Substitute (22) in (21), and ${\tilde{e}}_{i,l}$ is expressed as
(23)$${\tilde{e}}_{i,l}=r_{i}^{\left(l-1\right)}{}^{}-y_{i}^{\left(l-1\right)}\;,\;i\in \left\{1,\ldots ,\;p\right\},\;l\in \left\{1,2,\ldots ,\gamma_{i}\right\}$$Differentiate (23) w.r.t *t*, to get
(24)$${\dot{\tilde{e}}}_{i,l}=r_{i}^{\left(l\right)}{}^{}-y_{i}^{\left(l\right)}={\tilde{e}}_{i,\;l+1}\;,\;$$
then, ${\dot{\tilde{e}}}_{i,l}$ are expressed as,
(25)$$\{\begin{array}{c} \begin{array}{c} {\dot{\tilde{e}}}_{i,1}={\tilde{e}}_{i,2},\;\\ {\dot{\tilde{e}}}_{i,2}={\tilde{e}}_{i,3},\; \end{array}\;\\ \begin{array}{c} \vdots \;\\ {\dot{\tilde{e}}}_{i,\gamma_{i}}=r_{i}^{\left(\gamma_{i}\right)}-y_{i}^{\left(\gamma_{i}\right)}=r_{i}^{\left(\gamma_{i}\right)}-{\dot{\xi }}_{i,\gamma_{i}}\;,\;i\in \left\{1,\ldots ,\;p\right\}\; \end{array} \end{array}\;$$This together with (6b) gives,
(26)$$\{\begin{array}{c} \begin{array}{c} {\dot{\tilde{e}}}_{i,1}={\tilde{e}}_{i,2},\;\\ {\dot{\tilde{e}}}_{i,2}={\tilde{e}}_{i,3},\; \end{array}\;\\ \begin{array}{c} \vdots \;\\ {\dot{\tilde{e}}}_{i,\gamma_{i}}=r_{i}^{\left(\gamma_{i}\right)}{}^{}-\left(F_{i}^{\prime }+b_{i,i}u_{i}\right),\;i\in \left\{1,\ldots ,\;p\right\} \end{array}\; \end{array}$$Substituting (18) in (26), one obtains
(27)$$\{\begin{array}{c} \begin{array}{c} {\dot{\tilde{e}}}_{i,1}={\tilde{e}}_{i,2},\;\\ {\dot{\tilde{e}}}_{i,2}={\tilde{e}}_{i,3},\; \end{array}\;\\ \begin{array}{c} \vdots \;\\ {\dot{\tilde{e}}}_{i,\gamma_{i}}=r_{i}^{\left(\gamma_{i}\right)}-b_{i,i}v_{i}+{\hat{\xi }}_{i,\;\gamma_{i}+1}-F_{i}^{\prime },\;i\in \left\{1,\ldots ,\;p\right\} \end{array}\; \end{array}$$It follows from Theorem 1 that,
(28)$$\{\begin{array}{c} \begin{array}{c} {\dot{\tilde{e}}}_{i,1}={\tilde{e}}_{i,2},\;\\ {\dot{\tilde{e}}}_{i,2}={\tilde{e}}_{i,3},\; \end{array}\;\\ \begin{array}{c} \vdots \;\\ {\dot{\tilde{e}}}_{i,\gamma_{i}}=r_{i}^{\left(\gamma_{i}\right)}-b_{i,i}v_{i},\;i\in \left\{1,\ldots ,\;p\right\} \end{array}\; \end{array}$$The tracking errors dynamics of the (28) with the control signal $v_{i}$ of (19) yields the dynamics for the closed-loop errors given as,
(29)$$\{\begin{array}{c} \begin{array}{c} {\dot{\tilde{e}}}_{i,1}={\tilde{e}}_{i,2},\;\\ {\dot{\tilde{e}}}_{i,2}={\tilde{e}}_{i,3},\; \end{array}\;\\ \begin{array}{c} \vdots \;\\ {\dot{\tilde{e}}}_{i,\gamma_{i}}=-b_{i,i}{𝓀}_{i,1}\left({\tilde{e}}_{i,1}\right){\tilde{e}}_{i,1}-b_{i,i}{𝓀}_{i,2}\left({\tilde{e}}_{i,2}\right){\tilde{e}}_{i,2}-\ldots -b_{i,i}{𝓀}_{i,\gamma_{i}}\left({\tilde{e}}_{i,\gamma_{i}}\right){\tilde{e}}_{i,\gamma_{i}},\;i\in \left\{1,\ldots ,\;p\right\} \end{array} \end{array}$$The above dynamics of (29) is written in matrix notation,
(30)$${\dot{\tilde{e}}}_{i}=A_{i}{\tilde{e}}_{i},\;i\in \left\{1,\ldots ,\;p\right\}$$
where
(31)$$A_{i}=\{\begin{array}{cccccc} 0 & 1 & 0 & \ldots & 0 & 0\\ 0 & 0 & 1 & \ldots & 0 & 0\\ \vdots & \ldots & \ldots & \ldots & \vdots & \vdots \\ 0 & 0 & 0 & \ldots & 1 & 0\\ 0 & 0 & 0 & \ldots & 0 & 1\\ -{\tilde{𝓀}}_{i,1}\left({\tilde{e}}_{i,1}\right) & -{\tilde{𝓀}}_{i,2}\left({\tilde{e}}_{i,2}\right) & -{\tilde{𝓀}}_{i,3}\left({\tilde{e}}_{i,3}\right) & \ldots & -{\tilde{𝓀}}_{i,\gamma_{i}-1}\left({\tilde{e}}_{i,\gamma_{i}-1}\right) & -{\tilde{𝓀}}_{i,\gamma_{i}}\left({\tilde{e}}_{i,\gamma_{i}}\right) \end{array}$$
where ${\tilde{𝓀}}_{i,1}=b_{i,i}{𝓀}_{i,l}$, $\;l\in \left\{1,2,\ldots ,\gamma_{i}\right\},\;i\in \left\{1,\ldots ,\;p\right\}$, and ${\tilde{e}}_{i}=$ ${\left({\tilde{e}}_{i,1},{\tilde{e}}_{i,2},\ldots ,{\tilde{e}}_{i,\;\gamma_{i}}\right)}^{T}$. The characteristic polynomial of $A_{i}$ is given by,
(32)$$\left|\lambda I-A_{i}\right|=\lambda^{\gamma_{i}}+{\tilde{𝓀}}_{i,\gamma_{i}}\left({\tilde{e}}_{i,\gamma_{i}}\right)\lambda^{\gamma_{i}-1}+{\tilde{𝓀}}_{i,\gamma_{i}-1}\left({\tilde{e}}_{i,\gamma_{i}-1}\right)\lambda^{\gamma_{i}-2}+\ldots +{\tilde{𝓀}}_{i,1}\left({\tilde{e}}_{i,1}\right)$$The NLSEF controller adopted in this paper is utilizing the $fal_{i,l}\left(\cdot \right)$ expressed by (8) and is redrafted in terms of ${𝓀}_{i,l}\left(\cdot \right)$ as follows,
(33)$$fal_{i,l}\left({\tilde{e}}_{i,l},\alpha ,\delta \right)={𝓀}_{i,l}\left({\tilde{e}}_{i,l},\alpha_{i,l},\delta_{i,l}\right){\tilde{e}}_{i,l}\;$$
with $l\in \left\{1,2,\ldots ,\gamma_{i}\right\}$, i∈{1,…, p}, where
(34)$${𝓀}_{i,l}\left({\tilde{e}}_{i,l},\alpha_{i,l},\delta_{i,l}\right)=\{\begin{array}{cc} \frac{1}{\delta^{1-\alpha_{i,l}}} & \left|{\tilde{e}}_{i,l}\right|\leq \delta_{i,l}\\ {\left|{\tilde{e}}_{i,l}\right|}^{\alpha_{i,l}-1} & \left|{\tilde{e}}_{i,l}\right|\geq \delta_{i,l} \end{array}$$
is an even positive function. The coefficients ($\alpha_{i,l},\;\delta_{i,l})$ of (34) and $b_{i,i}$, i∈{1,…, p} are chosen to guarantee that the eigenvalues of (31) lie in the left-half plane, i.e., it is a Hurwitz polynomial. □ 

Moreover, the closed-loop stability (Observer/controller/Plant) for the proposed IADRC can be proved using the ISS (input-state-stability) framework or Lyapunov tools as in [52,53,54].

### 4.4. Relative Gain Array and Decentralized Control System Design

The RGA is very important in practical applications as it measures the interactions between different subsystems in MIMO systems. It can be described by,
$$\mu_{ij}=g_{ij}{\left[G\left(x\right)\right]}_{ji}^{-1},\;i,\;j=1,\;2$$

Depending on the values and the signs of the elements of the RGA, we can decide the suitable pairings between inputs and outputs. However, An ADRC approach makes perfectly good sense in the context of decentralized control and limited availability of state measurements for each agent in charge of control inputs *u*_1_ and *u*_2_. A first possibility is to regard the following scenario: *u*_1_ is in charge of controlling *y*_1_ and *u*_2_ is in charge of controlling *y*_2_. Another scenario is possible which is *u*_1_ is in charge of controlling *y*_2_ and *u*_2_ is in charge of controlling *y*_1_ as long as these scenarios are not violating the pairing rules which are: (a) to minimize interaction, variables with relative gains closest to 1 should be paired, (b) variables with negative gains should not be paired for control, and (c) relative gains of greater than 5 usually imply severe loop interaction. In our work we have chosen *u*_1_ to be in charge of controlling *y*_1_ and *u*_2_ to be in charge of controlling *y*_2_ and the effect of *u*_2_ on the 1st subsystem is considered as part of the generalized disturbance which will be estimated and canceled by the NHOESO. The same is applied to the 2nd subsystem of the MIMO system. To limit the size of the paper we have omitted the detailed calculations of the RGA and its role in the decentralized feedback control system design.

## 5. Numerical Simulations

To validate the proposed scheme performance for nonlinear MIMO system, we examine the following nonlinear multi-input-multi-output system,
(35){{ξ˙1,1=ξ1,2, ξ˙1,2=f1(ξ,η,w1)+g1,1(t)u1+g1,2(t)u2,y1=ξ1,1, {ξ˙2,1=ξ2,2, ξ˙2,2=f2(ξ,η,w2)+g2,1(t)u1+g2,2(t)u2, y2=ξ2,1, η˙=ξ1,2+ξ2,1+sin(η)+sin(t) 
where y1, y2 are the outputs, u1, u2 are inputs, η∈ℝ is the internal state of (35), ξ={ξ1,1, ξ1,2,ξ2,1,ξ2,2}∈ℝ4 is the external state vector. The variables y1, y2, u1, u2, w1,w2 belong to ℝ, and the functions f and g are given as,
(36){f1=ξ1,1+ξ2,1+η+sin(ξ1,2+ξ2,2)w1, f2=ξ1,2+ξ2,2+η+sin(ξ1,1+ξ2,1)w2, g1,1=1+110sin(t),g1,2=1+110cos(t), g2,1=1+1102−t,g2,2=−1 

Consider that the reference signals $r_{1},r_{2}$ and the external disturbance $w_{1},w_{2}$ are given as: $r_{1}=sin\left(t\right),\;r_{2}=cos\left(t\right),w_{2}=2^{-t}cos\left(t\right),w_{1}=1+sin\left(t\right).$ The states initial values are assumed as: $\left(\xi_{1,1},\xi_{1,2},\xi_{2,1},\xi_{2,2},\eta \right)=\left(0.5,\;0.5,\;1,\;1,0\right)$. Two ADRC scenarios will be adopted in the simulations. The dissimilarity among them is the kind of extended state observer (ESO) adopted to observe the system’s states and the generalized disturbance. The *first configuration* is the Conventional ADRC (CADRC), which involves a conventional TD described by [29],
(37)$$\{\begin{array}{c} {\dot{r}}_{i,1}=r_{i,2},\;\\ {\dot{r}}_{i,2}=-R_{i}sign\left(r_{i,1}-r_{i}+\frac{r_{i,2}\left|r_{i,2}\right|}{2R_{i}}\right),i\in \left\{1,2\right\} \end{array}$$
where $R_{i}$ i∈{1,2} is a design parameter.

The control law is designed based on the *fal*-function given as,
(38)$$\{\begin{array}{c} u_{1}=k_{1,1}fal\left({\tilde{e}}_{1,1},\alpha_{1,1},\delta_{1,1}\right)+k_{1,2}fal\left({\tilde{e}}_{1,2},\alpha_{1,2},\delta_{1,2}\right)-{\hat{\xi }}_{1,3}/b_{1,1}\\ u_{2}=k_{2,1}fal\left({\tilde{e}}_{2,1},\alpha_{2,1},\delta_{2,1}\right)+k_{2,2}fal\left({\tilde{e}}_{2,2},\alpha_{2,2},\delta_{2,2}\right)-{\hat{\xi }}_{2,3}/b_{2,2} \end{array}$$
where ${\tilde{e}}_{i,j}=r_{i,j}-{\hat{\xi }}_{i,j}\;\mathrm{for}\;i,j\in \left\{1,2\right\}$ is the tracking error, $k_{i,j},\;\alpha_{i,j},\delta_{i,j}\;i,j\in \left\{1,2\right\}$ are design coefficients of the *fal*-based control law, and an LESO given as
(39)$$\{\begin{array}{c} {\dot{\hat{\xi }}}_{i,1}={\hat{\xi }}_{i,2}+3\omega_{i,0}\left(y_{i}-{\hat{\xi }}_{i,1}\right),\;\\ {\dot{\hat{\xi }}}_{i,2}={\hat{\xi }}_{i,3}+b_{i,i}\;u_{i}+3\omega_{o,i}^{2}\left(y_{i}-{\hat{\xi }}_{i,1}\right),\;\\ {\dot{\hat{\xi }}}_{i,3}=\omega_{o,i}^{3}\;\left(y_{i}-{\hat{\xi }}_{i,1}\right),i\in \left\{1,2\right\}\; \end{array}$$
where $\;{\left({\hat{\xi }}_{i,1},{\hat{\xi }}_{i,2},{\hat{\xi }}_{i,3}\right)}^{T}$, i∈{1,2} are the estimated states and $\omega_{o,\;i}$, i∈{1,2} is the LESO bandwidth of the *i*th sub-system.

The IADRC is the *second configuration*, which involves of a conventional TD (35), a *fal*-based control law (36), and an NHOESO proposed as,
(40)$$\left\{ \begin{array}{c} {\dot{\hat{\xi }}}_{i,1}={\hat{\xi }}_{i,2}+a_{i,1}\omega_{o,\;i}{ℊ}_{i}\left(y_{i}-{\hat{\xi }}_{i,1}\right)\;\\ {\dot{\hat{\xi }}}_{i,2}={\hat{\xi }}_{i,3}+b_{i,i}\;u_{i}+a_{i,2}\omega_{o,i}^{2}\;{ℊ}_{i}\left(y_{i}-{\hat{\xi }}_{i,1}\right),\\ {\dot{\hat{\xi }}}_{i,3}={\hat{\xi }}_{i,4}+a_{i,3}\omega_{o,i}^{3}\;{ℊ}_{i}\left(y_{i}-{\hat{\xi }}_{i,1}\right),\;\\ {\dot{\hat{\xi }}}_{i,4}=a_{i,4}\omega_{o,i}^{4}\;{ℊ}_{i}\left(y_{i}-{\hat{\xi }}_{i,1}\right),\;i\in \left\{1,2\right\}\; \end{array}\right.$$
where the vectors ${\left({\hat{\xi }}_{i,1},{\hat{\xi }}_{i,2}{\hat{\xi }}_{i,3},{\hat{\xi }}_{i,4}\right)}^{T}$, i∈{1,2} are the estimated states $a_{i,j}$ is the associated design parameter, and $\omega_{o,\;i}$, i∈{1,2} is the NHOESO bandwidth the *i*th sub-system.

The function ℊi:ℝ→ℝ is proposed as in [50],
(41)ℊi(e)=Ki,α|e|αisign(e)+Ki,β|e|βie
where Ki,α,Ki,β,αi and βi are the positive design parameters.

### 5.1. The Results of the Proposed Scheme

The suggested decentralized control scheme based on IADRC for the MIMO system (33) is tested for reference tracking of $r_{1}$ and $r_{2}$. The suggested control signals $u_{1}$ and $u_{2}$ are same as in (36) but with saturation, i.e., they are reformulated as
(42)$$\{\begin{array}{c} u_{1}=sat\left(k_{1,1}fal\left({\tilde{e}}_{1,1},\alpha_{1,1},\delta_{1,1}\right)+k_{1,2}fal\left({\tilde{e}}_{1,2},\alpha_{1,2},\delta_{1,2}\right)-{\hat{\xi }}_{1,3}/b_{1,1},\delta_{1}\right)\\ u_{2}=sat\left(k_{2,1}fal\left({\tilde{e}}_{2,1},\alpha_{2,1},\delta_{2,1}\right)+k_{2,2}fal\left({\tilde{e}}_{2,2},\alpha_{2,2},\delta_{2,2}\right)-{\hat{\xi }}_{2,3}/b_{2,2},\delta_{2}\right) \end{array}$$
where $\delta_{i}$, i∈{1,2} is a design parameter and sat(u,δ) is expressed as
(43)$$sat\left(u,\delta \right)=\{\begin{array}{c} \begin{array}{cc} \delta & u\geq \delta \end{array}\;\\ \begin{array}{cc} u & -\delta <u<\delta \end{array}\\ \begin{array}{cc} -\delta & u\leq -\delta \end{array}\; \end{array}$$

The two CADRC and the IADRC configurations will be utilized in the decentralized control scheme. The control signals $u_{i},$ i∈{1,2} indicated in Figure 4 is resulted from control law based on the *fal*-functions given in (42). The desired transient trajectories $\;{\left(r_{1,1},r_{1,2}\right)}^{T}$, and ${\left(r_{2,1},r_{2,2}\right)}^{T}$ are the outputs of the TD defined in (37). In Table 1 and Table 2, the parameters of the IADRC and CADRC structures are listed, respectively, where they are tuned using Genetic algorithm (GA) to minimize the multi-objective performance index (MOPI) which is a combination of the ITAE, ISU. The resulting performance measures of the numerical simulation are listed in Table 3, where $\mathrm{ITAE}=\int_{0}^{t_{f}}t\left|y-r\right|dt$ is the time absolute error integration, $\mathrm{ISU}=\int_{0}^{t_{f}}v^{2}\;dt$ is the integrated squared control signal v, while the output response curves for the proposed decentralized ADRC scheme with the two configurations are shown in Figure 5 and Figure 6, respectively.

As shown in Table 3, the decrease in the performance indices ISU and ITAE for the two subsystems in the IADRC is obvious in correlation to CADRC configuration. Furthermore, the signals $u_{1}$ and $u_{2}$ with fewer chatterings have been generated by the IADRC configuration in correlation with their partners in the CADRC setup. In the case of the IADRC, a better output tracking has been obtained as compared to CADRC configuration, explicitly during the transient time response, where the two setups have totally constricted the impact of the state coupling for every subsystem, the exogenous disturbances $w_{1}$ and $w_{2}$, and the time-varying input gains $b_{1,1}$, $b_{1,2}$, $b_{2,1}$, and $b_{2,2}$ on the output of each channel. The transient response of the outputs $y_{1}$ and $y_{2}$ due to the desired inputs $r_{1}$ and $r_{2}$ successively imposed to (35) are displayed in Figure 7 and Figure 8. It can be concluded that the decoupling necessity expressed in the statement of the problem is totally fulfilled with a flat output for every output subsystem. The suggested configuration transformed the nonlinear MIMO system of (35) into two isolated SISO subsystems.

### 5.2. Discussion

To control MIMO uncertain nonlinear systems, couplings between various individual subsystems are regarded as the most substantial intricacy. In this manner, it is important to foster a control method that is both straightforward and vigorous. A decentralized control scheme, which utilizes the IADRC due to its model-free independence and robustness highlights, has been recommended in this work. The nonlinear couplings together with the system uncertainty have been treated within the generalized disturbances $\xi_{1,3}\;\mathrm{and}\;\xi_{2,3}$ that have been observed and canceled via the IADRC design in an online manner. In the ADRC-based decentralized control configuration, assume that the generalized total disturbances up unsettling influences $\xi_{1,3}\;\mathrm{and}\;\xi_{2,3}$ will have more terms to be canceled, for example, undesirable control input, external disturbance, system uncertainties, undesirable system dynamics, and so forth, the correctness of the ADRC-based decentralized control configuration is decreased further. This decrease is obvious in Figure 5c,d and Figure 6c,d where huge chatterings are found in the efforts $u_{1},\;u_{2}$ and the generalized disturbance $\xi_{1,3},\;\xi_{2,3}$ at the very beginning of the simulations and disappeared rapidly due to the previously mentioned reasons. Finally, it is of concern to mention the advantage of the proposed decentralized control scheme is then aimed at reducing a complex process to a possibly linear perturbed plant affected by a total disturbance term, which is easy to control by means of a linear output feedback control law. While the disadvantage of the proposed decentralized control scheme is that in application, the ESO requires little information from the system in its estimation of unmeasured states, uncertainties and exogenous disturbances, and is thus frequently preferred in the design of feedback controllers.

## 6. Conclusions

In this paper, an IADRC was proposed to control multi-input-multi-output uncertain nonlinear systems. The proposed control configuration, i.e., the decentralized configuration used the ADRC technique due to its aforesaid superior feature. The proposed control scheme efficiently eliminated the input couplings, state couplings, exogenous disturbances, and system uncertainties via the NHOESO, which is the central part of the IADRC configuration. The simulations showed that the proposed decentralized control scheme with the IADRC technique converted the uncertain nonlinear MIMO system into a distinct multiple SISO linear time-invariant system with suitable state feedback control law. Consequently, the IADRC-based decentralized scheme has a higher chance for practical implementation because of the simplicity in incorporating signals from different subsystems as a portion of the estimated generalized total disturbances. Moreover, it can be inferred that the performance of the proposed IADRC-based decentralized control scheme is remarkably better than its counterpart of the CADRC-based decentralized control scheme regarding output tracking, control energy, and chattering as indicated by Table 3. A possible future work is to implement the proposed control configuration on a real MIMO nonlinear testbed and to compare practical results with that of the simulations presented in this paper.

## Figures and Tables

**Figure 1 entropy-23-01487-f001:**
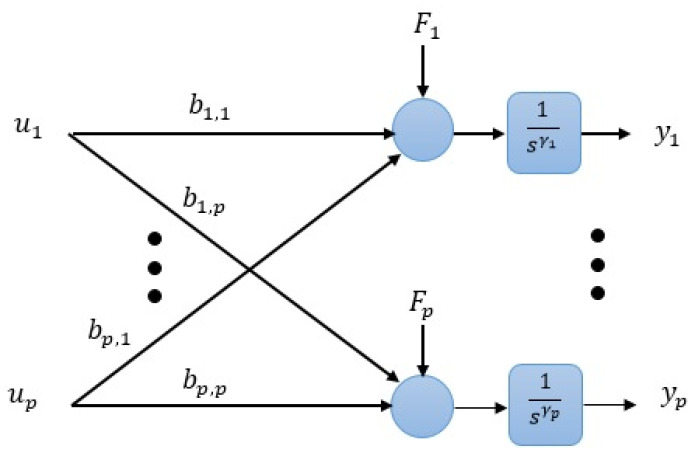
The diagram of the nonlinear MIMO system based on (2) and (3).

**Figure 2 entropy-23-01487-f002:**
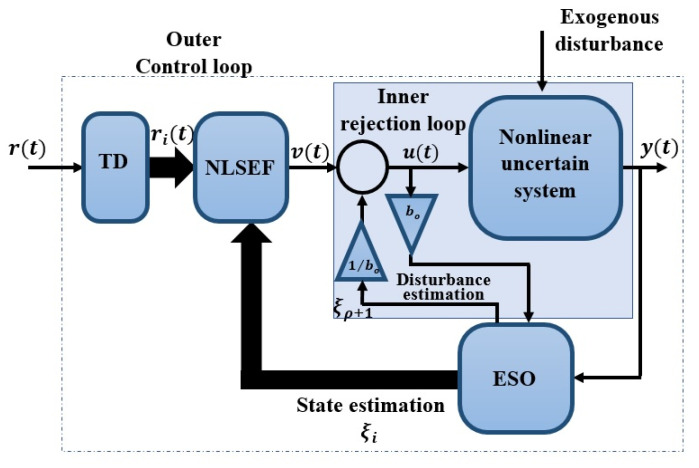
Structure of the conventional SISO ADRC configuration, i∈{1,2,…,ρ}, ρ is the nonlinear system relative degree.

**Figure 3 entropy-23-01487-f003:**
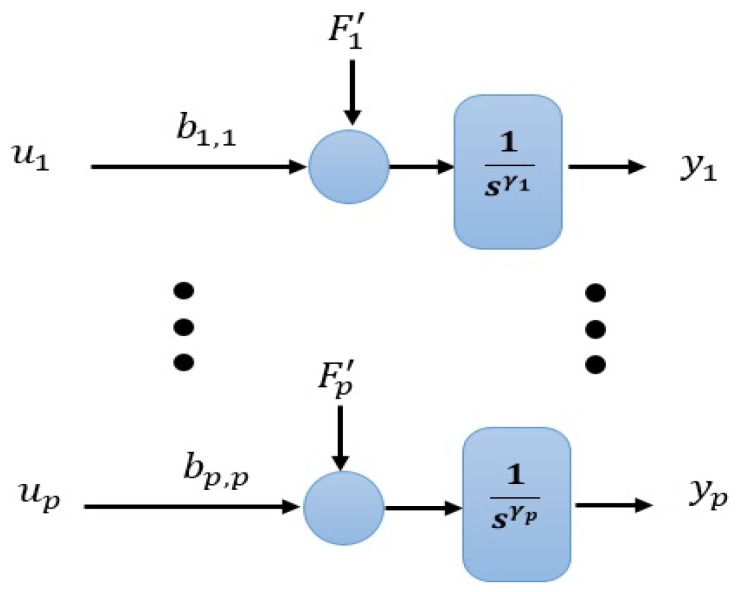
The diagram of the uncertain nonlinear MIMO system of (6).

**Figure 4 entropy-23-01487-f004:**
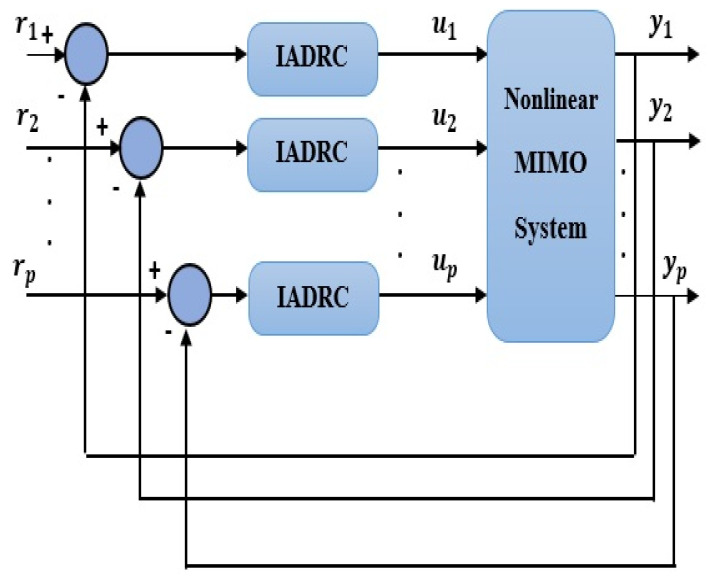
The proposed IADRC-based decentralized control scheme.

**Figure 5 entropy-23-01487-f005:**
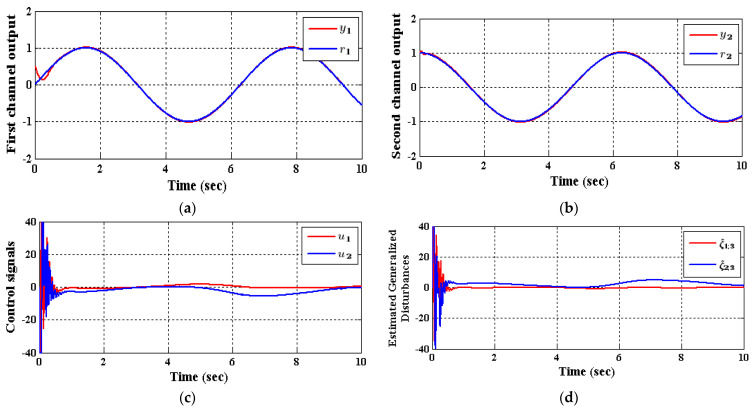
The response of (35) employing CADRC, (**a**) $y_{1}$, (**b**) $y_{2}$, (**c**) $u_{1}$ and $u_{2}$, (**d**) observed disturbances ${\hat{\xi }}_{1,3}$ and ${\hat{\xi }}_{2,3}$.

**Figure 6 entropy-23-01487-f006:**
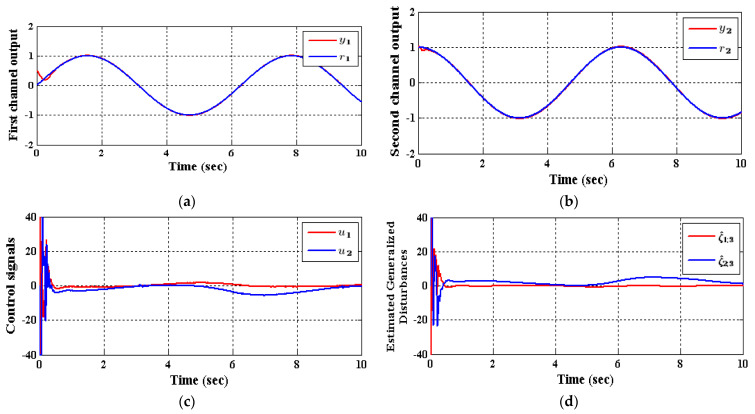
The response of (35) utilizing IADRC, (**a**) $y_{1}$, (**b**) $y_{2}$, (**c**) $u_{1}$ and $u_{2}$, (**d**) observed disturbances ${\hat{\xi }}_{1,3}$ and ${\hat{\xi }}_{2,3}$.

**Figure 7 entropy-23-01487-f007:**
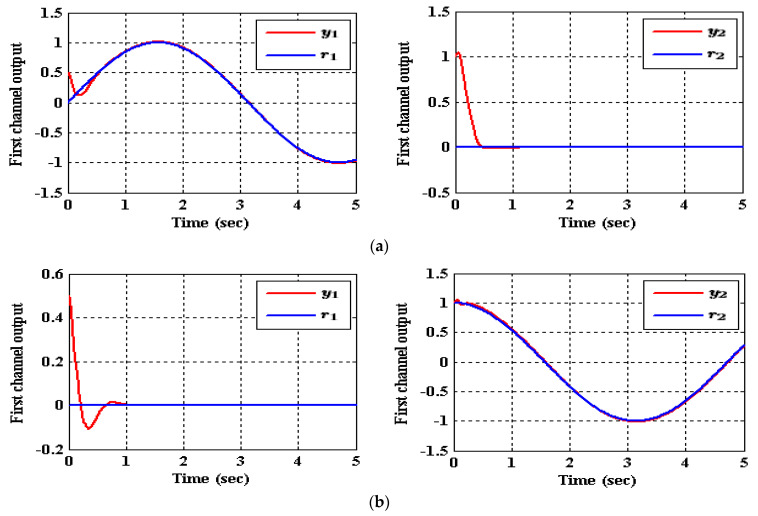
The tracking of (35) for reference signals $r_{1}$ and $r_{2}$ employing CADRC, (**a**) $\left(r_{1},r_{2}\right)=\left(sin\left(t\right),0\right)$, (**b**) $\left(r_{1},r_{2}\right)=\left(0,cos\left(t\right)\right)$.

**Figure 8 entropy-23-01487-f008:**
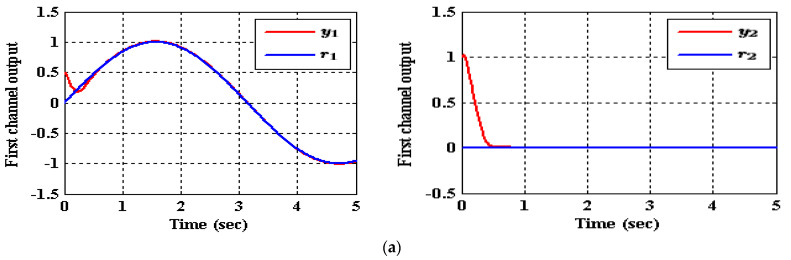
The tracking of (35) for reference signals $r_{1}$ and $r_{2}$ utilizing IADRC, (**a**) $\left(r_{1},r_{2}\right)=\left(sin\left(t\right),0\right)$, (**b**) $\left(r_{1},r_{2}\right)=\left(0,cos\left(t\right)\right)$.

**Table 1 entropy-23-01487-t001:** The coefficients values of CADRC.

Unit	1st Channel		2nd Channel	
	Coefficient	Value	Coefficient	Value
TD	$R_{1}$	92.2713	$R_{2}$	88.4423
LESO	$\omega_{o,1}$	68.3308	$\omega_{o,2}$	53.1690
	$b_{1,1}$	1.0000	$b_{2,2}$	−1.0000
fal-based Control law	$\delta_{1,1}$	0.0010	$\delta_{2,1}$	0.1445
	$\delta_{1,2}$	0.2834	$\delta_{2,2}$	0.7346
	$b_{1,1}$	1.0000	$b_{2,2}$	−1.0000
	$\alpha_{1,1}$	0.1629	$\alpha_{2,1}$	0.0273
	$\alpha_{1,2}$	0.7946	$\alpha_{2,2}$	0.9375
	$k_{1,1}$	12.8015	$k_{2,1}$	18.3095
	$k_{1,2}$	11.2999	$k_{2,2}$	19.5267
	$\delta_{1}$	40	$\delta_{2}$	40

**Table 2 entropy-23-01487-t002:** The coefficients values of IADRC.

Unit	1st Channel		2nd Channel	
	Coefficient	Value	Coefficient	Value
TD	$R_{1}$	155.2564	$R_{2}$	107.6494
NHOESO	$\omega_{o,1}$	94.9942	$\omega_{o,2}$	123.7601
	$b_{1,1}$	1.0000	$b_{2,2}$	−1.0000
	$a_{1,1}$	1.7315	$a_{2,1}$	3.6546
	$a_{1,2}$	5.0845	$a_{2,2}$	3.8128
	$a_{1,3}$	1.5151	$a_{2,3}$	2.0353
	$a_{1,4}$	1.1444 × 10^−6^	$a_{2,4}$	1.1230 × 10^−6^
	$K_{1,\alpha }$	0.8028	$K_{2,\alpha }$	0.5043
	$\alpha_{1}$	0.9300	$\alpha_{2}$	0.6982
	$K_{1,\beta }$	0.2381	$K_{2,\beta }$	0.8338
	$\beta_{1}$	0.6221	$\beta_{2}$	0.9534
fal-based Control law	$\delta_{1,1}$	0.1250	$\delta_{2,1}$	0.2510
	$\delta_{1,2}$	0.4163	$\delta_{2,2}$	0.4531
	$b_{1,1}$	1.0000	$b_{2,2}$	−1.0000
	$\alpha_{1,1}$	0.2750	$\alpha_{2,1}$	0.3312
	$\alpha_{1,2}$	0.7658	$\alpha_{2,2}$	0.2783
	$k_{1,1}$	25.6305	$k_{2,1}$	30.3227
	$k_{1,2}$	10.6899	$k_{2,2}$	20.2694
	$\delta_{1}$	40	$\delta_{2}$	40

**Table 3 entropy-23-01487-t003:** Performance of the decentralized ADRC scheme.

Performance Index	CADRC	IADRC	%Reduction
ITAE1	0.3890	0.3081	20.8%
ITAE2	0.6434	0.4600	28.5%
ISU1	181.5489	123.6903	31.9%
ISU2	302.3266	265.2197	12.3%

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
