# Peer review of "Anti-Disturbance Compensation-Based Nonlinear Control for a Class of MIMO Uncertain Nonlinear Systems"

_entropy, 2021, doi:10.3390/e23111487_

Round 1

Reviewer 1 Report

Authors devise a modified ADRC technique in MIMO nonlinear systems in distributed settings. 

The paper has some innovation and value addition but it needs some revision based on the following comments. 

1) The title is inappropriate. The term "Anti-disturbance" is misleading from the work in the paper. 
I suggest to replace the title by "Improved Active Disturbance Rejection Based Distributed Control for a Class of MIMO Nonlinear Systems". 

2) Authors should cite some of the recent results on Sliding Mode and higher order sliding mode Control which is a powerful robust control technique.  For instance, the following references include similar complex nonlinear systems with disturbance rejection.

1. " Sliding mode control for an underactuated slosh-container system using nonlinear model" , lnternational Journal of Advanced Mechatronic Systems. Vol.5, No.5, pp. 335-344, 2013.
2. "lmproved output-feedback second order sliding mode control design with implementation for underactuated slosh-container system having confined track length", IET Contro Theony and Applications, 11(8), pp. 1316-1323, 2017.

3) About equation (4), authors say that it is the modification of equation (1) but i feel due to the bifurcation of the last two terms in (4), it is one and the same equation as (1). Please clarify and write appropriately.

4) Through the different state assignments for $\xi$, it is not clear how equation (6-b) is achieved from (6-a). Please elaborate very clearly as this is the part that leads to the crucial decoupling being done further. 

5) Some statements in the introduction section are not lucid and can be improved for better readability. This is also true for further english polishing needed in the remaining of the paper. e.g. 
- on page 7, line 248, remove the acronym, LCL for linear control law. it is unnecessary. 
- remove the underline for the word large-scale systems in page 2, line 64.
and in the same sentence the use of an auxiliary verb 'is' is inappropriate for the plural 'large-scale systems'. 
etc. 

Author Response

response to reviewer -1

Reviewer 2 Report

The current paper proposes proposes IADRC to control multi-input-multi-output uncertain nonlinear systems. The proposed theory is validated using simulations.

Comments to author:
- The paper quality can be improved by adding scheme block diagram of the control structure with the proposed ADRC controller.
- How the authors ensure and guarantee the stability of the control system structure?
- Please add more details of how the theory is applied in simulation setup.
- How the parameters of the ADRC algorithm were chosen?
- The authors could add a paragraph with the advantages and the disadvantages of the proposed ADRC algorithm.
- The state of the art should be improved with more references based on other controller type that can be applied in cascade fuzzy logic control, maybe the author could add the following publications:
o Hybrid data-driven fuzzy active disturbance rejection control for tower crane systems, European Journal of Control, vol. 58, pp. 373-387, 2021. 
o On model free adaptive control and its stability analysis, IEEE Transactions on Automatic Control, doi 10.1109/TAC.2019.2894586, pp. 1–14, 2019. 
- Please add the graph axis with the units of measurement for all figures.

Author Response

response to reviewer -2

Reviewer 3 Report

In  this paper  a control scheme  is proposed based on the decentralized principle in which  the input couplings for the uncertain nonlinear multi-inputs-multi outputs system is first resolved ,converting it into decoupled linear time-invariant single -output systems .

The simulations  showed  that the proposed decentralized  control scheme has a higher  chance for practical implementation.

The paper is unitary , well structured and the obtained results are interesting.

In conclusion , I consider that the paper contain an original research in the area and I recommend publication.

Author Response

response to reviewer -3

Round 2

Reviewer 2 Report

The current paper has been seriously improved, the authors answered to all my concerns. From my point of view the paper can be accepted as contribution in Entropy journal.